# Interclutch variability in egg characteristics in two species of rail: Is maternal identity encoded in eggshell patterns?

**Emily W. Johnson, Susan B. McRae** [ID] *

Department of Biology, East Carolina University, Greenville, North Carolina, United States of America

* mcraes@ecu.edu

## Abstract

Maternal signatures are present in the eggs of some birds, but quantifying interclutch variability within populations remains challenging. Maternal assignment of eggs with distinctive appearances could be used to non-invasively identify renesting females, including hens returning among years, as well as to identify cases of conspecific brood parasitism. We explored whether King Rail (*Rallus elegans*) eggs with shared maternity could be matched based on eggshell pattern. We used NaturePatternMatch (NPM) software to match egg images taken in the field in conjunction with spatial and temporal data on nests. Since we had only a small number of marked breeders, we analyzed similar clutch images from a study of Eurasian Common Moorhens (*Gallinula chloropus chloropus*) with color-banded breeders for which parentage at many nests had been verified genetically to validate the method. We ran 66 King Rail clutches ($n = 338$ eggs) and 58 Common Moorhen clutches ($n = 364$ eggs) through NPM. We performed non-metric multidimensional scaling and permutational analysis of variance using the best egg match output from NPM. We also explored whether eggs could be grouped by clutch using a combination of egg dimensions and pattern data derived from NPM using linear discriminant analyses. We then scrutinized specific matches returned by NPM for King Rail eggs to determine whether multiple matches between the same clutches might reveal maternity among nests and inform our understanding of female laying behavior. To do this, we ran separate NPM analyses for clutches photographed over several years from two spatially distant parts of the site. With these narrower datasets, we were able to identify four instances where hens likely returned to breed among years, four likely cases of conspecific brood parasitism, and a within-season re-nesting attempt. Thus, the matching output was helpful in identifying congruent egg patterns among clutches when used in conjunction with spatial and temporal data, revealing previously unrecognized site fidelity, within-season movements, and reproductive interference by breeding females. Egg pattern data in combination with nest mapping can be used to inform our understanding of female reproductive effort, success, and longevity in King Rails. These methods may also be applied to other secretive birds and species of conservation concern.

**Data Availability Statement:** All data files and R code are available from the Data Dryad database (https://doi.org/10.5061/dryad.6q573n60j).

**Funding:** This study was conducted with support from the U.S. Fish and Wildlife Service, Refuge

System Inventory and Monitoring program through a Piedmont South Atlantic Coast Cooperative Ecosystems Studies Unit (http://www.cesu.psu.edu/unit_portals/PSAC_portal.htm) agreement to SBM (F19AC00629), and an E. Alexander Bergstrom Memorial Research Award from the Association of Field Ornithologists (https://afonet.org/grants-awards/bergstrom/) to EWJ. The funders had no role in study design, data collection and analysis, decision to publish, or preparation of the manuscript.

**Competing interests:** The authors have declared that no competing interests exist.

# Introduction

Since the 1800s, researchers and bird enthusiasts have pondered the question of why some species lay decorated eggs and what purpose this could serve [1]. Egg shapes vary from elliptical to spherical, ground colors range from white to bright blue to brown, and eggshells can be heavily patterned with additional pigments [2–5]. Alfred Russel Wallace was the first to suggest that variation in egg coloration represented the general fitness and health of the hen and that egg color and patterning were only constrained by predation [1]. The adaptation of pigment patterns in eggs is driven by natural selection with predation selecting for crypsis through patterns adapted to nest substrates [6]. Various hypotheses have additionally been proposed to explain variation in egg pigmentation within and between species, including thermoregulation of the embryo [6], enhanced shell strength [7], and antimicrobial properties of pigments [8].

In birds that lay maculated or spotted eggs, pattern 'signatures', unique or distinctive sets of characteristics, can develop that potentially allow a breeder to discriminate between its own eggs and those of others [2,9–12]. For example, maternal egg signatures can develop when conspecifics nest in close proximity and risk mistaking a neighbor's nest for their own, such as in colonial seabirds [13–15]. Maternal signatures can alternatively develop in response to brood parasitism [10]. Misdirected parental care is costly and reduces the host's ability to invest in its own young. A host that recognizes a foreign egg may reject it, avoiding the cost of raising the parasite's offspring [11,12]. Egg rejection requires hosts to be able to recognize their own eggs and distinguish those of the parasite. This has driven selection in many interspecific parasites for eggs that mimic those of the host, and in hosts for female-specific egg patterning [3,11]. Thus, brood parasitism can select for within-population egg variation, specifically reducing variation within clutches and increasing variation among clutches [16]. Here, hosts evolve egg signatures as a form of anti-parasite defense [9,13,14,17–19].

In conspecific brood parasitism (CBP), hosts and parasites are members of the same population [20], so selection for egg signatures will depend on the costs of parasitism to hosts and the benefits to parasites. Some populations experience both conspecific and interspecific parasitism. For example, colonial nesting *Ploceus* weaverbirds are parasitized by Diederik Cuckoos (*Chrysococcyx caprius*) as well as conspecifics, and individual females have developed distinctive female-specific coloring and patterning [9,17]. Exhibiting some of the most striking examples of egg signatures known, *Ploceus* weaverbird eggs vary greatly between females in ground color and patterning while showing strong consistency within clutch, facilitating assignment of maternal identity based on eggshell appearance [17,18].

Protoporphyrin, the pigment responsible for colors ranging from red to purple and brown, is typically expressed as spotting or speckling on the eggshell [21]. These maculations can range in size from pinprick speckles to spots and blotches covering large portions of the eggshell. Within species, protoporphyrin pigmented eggshell pattern variation has been used to determine maternity and discriminate between clutches in populations of birds such as the Herring Gull (*Larus argentatus*) [22], the Great Tit (*Parus major*) [23], *Ploceus* weaverbirds [9,17,18], and in cases of hosts of the interspecific parasite the Common Cuckoo (*Cuculus canorus*) [10].

Recent technological and statistical advances have furthered our ability to detect and compare variability in patterning and color. A variety of photo-based pattern detection and matching programs developed using information theory can identify and discriminate biological patterns in various contexts. For example, conservationists have used matching algorithms to identify individual newts [24], manta rays [25], whale sharks [26] and sea turtles [27] based on their distinctive skin and shell patterns for re-sighting purposes. Information theory-based

methods lack the subjectivity of earlier methods of pattern detection and recognition and take less time to complete [25,26].

NaturePatternMatch (hereafter, NPM) was developed specifically to evaluate matching characteristics of avian eggshells [10]. The program performs analysis of pattern features on high-quality digitized images of individual eggs and locates the best matching egg image among a sample of other eggs in the population. Images are converted to grayscale, and the program characterizes major pigment features on the eggshell surface to match it with the egg it most closely resembles [10,28]. Statistical tests such as principal component analyses, non-metric multidimensional scaling and cluster analysis can then be used to identify differences among differently patterned groups [10,15,29].

We investigated the existence of maternal egg signatures and whether they could be used to match eggs to clutches or hens in two species of rails that lay protoporphyrin maculated eggs, the King Rail and the Eurasian Common Moorhen. We theorized that both species evolved female-specific signatures in response to a history of CBP. Rails are well-known for exhibiting CBP and many species have developed anti-parasite defense strategies to mitigate the costs of parasitism such as nest abandonment, egg rejection and burial [30–34].

King Rail populations have declined significantly in the last decades [35]. There is an urgent need to be able to identify breeders in populations of conservation concern to monitor individual reproductive effort and success, but this remains a challenge because sightings of secretive marsh birds are rare even if adults are banded. Recognizing maternal egg signatures could provide an alternative means to non-invasively identify within-season renesting attempts by the same hen, and annual return rates of breeding females. Moreover, reliable maternal egg signatures could be used to detect incidences of reproductive interference (i.e. CBP) contributing to our understanding of social behavior.

To test whether NPM could be used to group eggs based on maternity, we studied a resident population of King Rails breeding in a coastal refuge and collected standardized clutch images over a period of 10 years. Due to the difficulty of catching and resighting King Rails, only a small proportion of breeders were marked. Therefore, we included data from a British population of the confamilial Common Moorhen (*G. c. chloropus*) with individually marked breeders for comparison. This was a well-studied population at Peakirk Waterfowl Gardens, Cambridgeshire, U.K., where identity of breeders was known from frequent re-sightings of marked birds at nests. There was also information on laying sequence and genetic parentage, including confirmed instances of CBP. The species are similar in size, developmental mode, parental care behavior, and both exhibit large within-population ranges in clutch size. A low rate of nest parasitism has been recorded in the King Rail population, and at least 10% of Common Moorhen nests were parasitized annually at Peakirk Waterfowl Gardens [36].

After initial tests of the ability of NPM to group eggs of each species according to clutch or hen, we investigated whether assignment success could be improved using linear discriminant analyses combining pattern data from NPM output in conjunction with a pigmentation index and egg dimensions. Finally, by comparing the matching output of NPM with the dates and mapped locations of King Rail nests, we identified instances of nesting by the same females both within and between years, as well as cases of CBP.

## Materials and methods

### Study sites and field data collection

**King Rails in coastal North Carolina.** This study was approved by the Institutional Animal Care and Use Committee of East Carolina University (Permit Number: D253c), under Federal Bird Banding Permit #23728 (SBM), State Wildlife Collection License

#19-SC00967 (SBM), U.S. Fish and Wildlife Service (Refuge Special Use Permit Number: 19004). All procedures were carried out in accordance with protocols outlined in these permits.

We studied a breeding population of King Rails at Mackay Island National Wildlife Refuge located on Knotts Island, NC, USA (36° 31′ N, 75° 58′ W). This population has been monitored annually during the breeding season (April-July) 2011–2020. King Rail pairs were located using auditory callback [37] and field teams found nests via intensive and systematic searching of appropriate marsh habitat on foot. Nests found before clutch completion were monitored daily during the laying period to determine laying sequence and afterward at least every 3 days to determine their final fate (depredated, hatched, deserted). When a new nest was located, the length and width of each egg was measured using dial calipers (± 0.1 mm; SPI). Each egg was given a unique number with a permanent marker (Sharpie). After clutch completion, a standardized photo was taken of the eggs laid out on a flat surface. The eggs were cleaned of debris (feces, mud, water droplets, vegetation) and placed on a custom-made board with egg-shaped holders covered in black velvet (to absorb light). The eggs were positioned such that they were not touching, in a consistent orientation and numerical order, and the identification numbers were not visible. Each picture included a size standard (ruler), a color standard (the same set of 10 plastic leg bands of different primary and secondary colors), and a label with the nest identity and date. Digital photographs were taken with a Nikon Coolpix P340 camera held approximately 12 inches directly above the clutch. The camera model varied in some years.

**Common Moorhens in Britain.** Common Moorhen data were obtained from a study of a resident population of wild birds nesting at Peakirk Waterfowl Gardens, Cambridgeshire, UK (52° 38′ 46″ N, 0° 16′ 16″ W) [36]. The moorhens in this population had small territories among the exhibits of captive waterfowl and were multiple-brooded. Breeding individuals were marked with unique color band combinations and were observed daily. DNA fingerprinting was also used to confirm parentage and to identify parasitic eggs [38]. This resulted in a dataset with many clutches of known genetic maternity. Clutches were photographed at completion of laying with a wing rule for scale. Eggs were placed on a flat surface, in a consistent orientation, spaced so that none were touching and no writing was visible. No color standard was used, but the same cloth background material was used for almost all photographs (Fig 1) [39]. The following measures were collected for each egg in the field: length (±0.1mm), width (±0.1mm), and mass (g, measured within 24 hrs of laying). Data were collected during the 1991–1993 breeding seasons (March-August) by SBM. For more details see McRae [36,39] and McRae and Burke [38].

## Pattern matching analysis

Individual egg images were matched to their correct clutches based on spot patterning using the program NaturePatternMatch v1.05. We ran photos of 66 King Rail clutches (338 eggs, 58 repaired using photoshop, 2 to 10 eggs per clutch, mean = 5.1) through NPM from all years except 2013 (all photos were of inadequate quality) and 2018 (few photos were available). Clutch photos that were out-of-focus, of low resolution, or in which all the eggs were dirty were also excluded. This led to the exclusion of 32 King Rail clutch photos. Some individual egg images had to be removed from the analysis because they were dirty or had visible marker numbering that could not be cropped out and repaired. In some cases, they were removed because the photograph was overexposed, shadowy, or the eggs had surface shine. We retained in the analysis partial clutches of as few as two eggs because adequate matching can occur even with only two eggs per clutch [10].

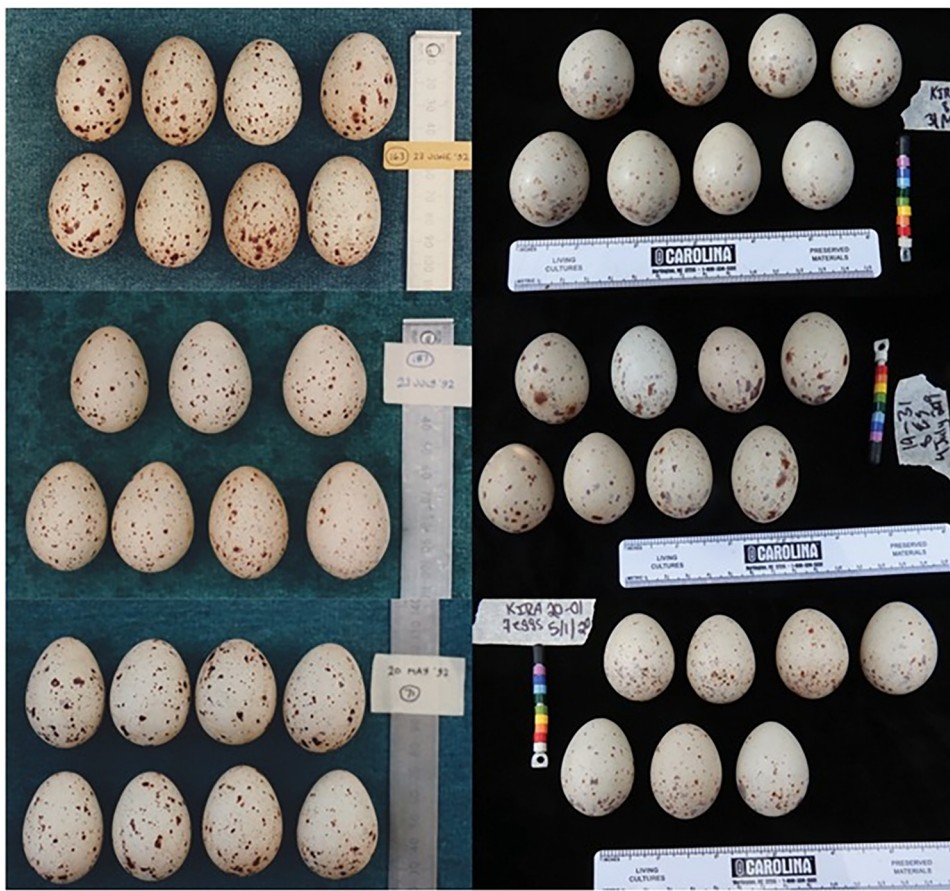

**Fig 1.** Representative clutch images of the Common Moorhen (left) and the King Rail (right) demonstrating the variation in patterning and color within and between clutches.

We used photographs of 58 Common Moorhen clutches taken between 1991–1993 (364 eggs, 5 to 10 eggs per clutch, mean = 6.3). The Common Moorhen clutches were photographed using Kodak 200 print film. We scanned 10.16x15.24 cm (4x6 inch) prints from 1991 and 1992 of Common Moorhen clutches ($n$ = 37) using a Canon Pixma TS3122 printer at 600 DPI (dots per inch). We scanned negatives of the clutch photos from 1993 using a Nikon Super CoolScan 9000 (ED 1.01) ($n$ = 21). The same criteria for Common Moorhen egg exclusions were used as were used for King Rails. Clutch images included in the analyses had to be in focus with good lighting and feature at least 5 clean eggs laid by a banded female. To increase the probability of having unrelated hens, and avoid possible effects of heritability of eggshell patterning, we excluded communal nests where hens were almost always first order relatives, such as mother and daughter [39]. Most hens in this population laid 2 to 4 clutches per season, and we included a small number of photographs of 2 or 3 clutches within a season from the same hen (1992 $n$ = 7, 1993 $n$ = 3). In one instance, we had suitable clutch photos from a female in all 3 years. For a subset of 7 females, we included clutches from two years.

We ran analyses of images for each year independently, excluding 2011 and 2015 (due to small sample sizes). For King Rails, we also ran all years combined, and all years together but divided by location within the refuge, denoted as North or South side. These two road-accessible areas of marsh where they bred were ~10km apart. A radiotelemetry study of King Rails found that tracked individuals did not move between the North and South sides of the refuge

[40]. This also allowed us to narrow the sample for comparison to proximate nests within- and among years, because a female may show site fidelity. For the moorhens, each year was run separately (due to differences in image capture and processing), and with all years combined. We extracted from the NPM output (SIFT files) the total number of features detected, the scale (size) of the largest feature, and the dominant orientation of the largest feature for each egg to be used in subsequent analyses.

## Quantifying pigment on eggshell surfaces

To estimate the amount of surface of each egg that was pigmented, egg images were rendered in black and white, and the proportion of black pixels was measured. First, each color image was converted to grayscale (8-bit) using ImageJ (NIH v. 1.52). The background was then removed and Bernsen thresholding, an algorithm which uses variance to calculate a threshold for each pixel [41], was performed. Bernsen thresholding was chosen because, upon visual inspection, black areas on egg images thresholded using this method more closely matched the pigmented areas on the color images when comparing to those thresholded using other methods.

A standardized 3.4 x 2.4 cm block in which a scaled oval was drawn was superimposed over each egg image. The oval was centered at the edge of the large pole of the egg and fit within both the narrowest and shortest eggs in the analysis, ensuring that the same total area was measured on every egg. This was done to capture the greatest area on all eggs, while also measuring the area of the egg where pigment was greatest (large pole). Bernsen thresholding produces a ring of black pixels around the egg, but placement of the scaled oval just within the black outline ensured that the pixels creating the black outline around the egg were not being counted. The number of black pixels within the oval was measured using the histogram tool. The number of white pixels within the oval was also measured, enabling the expression of the proportion of black pixels as a percentage.

## Statistical analyses

**Non-metric multidimensional scaling.** All analyses were performed using R version 4.0.2 [42]. To investigate how often eggs were matched with an egg of the correct clutch based on comparisons of clutches both within and between years, we conducted non-metric multidimensional scaling (NMDS). Though the goal of this study was to determine if females have maternally distinct eggs, NPM identifies the best pair-wise match between individual egg images; it does not specify the correct hen or match to clutch. For this reason, we used NPM to find best matches and then calculated the proportion of within clutch matches for both species. To determine if variation in patterning existed among clutches and hens, we performed Permutational Multivariate Analysis of Variance (PERMANOVA) for both species.

**Linear discriminant analysis of egg characteristics.** Eggs laid by different hens vary in size and shape as well. To incorporate this variation, we performed a linear discriminant analysis to determine if eggs of each species could be grouped by clutch or hen using a combination of eggshell pattern characteristics and egg dimensions. Exclusions were made due to missing data ($n$ = 27 King Rail eggs, $n$ = 3 Common Moorhen eggs) leaving 337 King Rail eggs and 361 Common Moorhen eggs in the analysis. The grouping variable was clutch for King Rails, and clutch or hen for the Common Moorhens. Both datasets were also verified for among year differences by using year as a grouping variable. The covariates included were egg length (mm), egg width (mm), egg mass (g, Common Moorhens only), percent black pixels, number of features, scale of the largest feature, and dominant orientation of the largest feature.

## Spatial and temporal analysis of King Rail matching results

Maps of all King Rail nests for which we had usable clutch photographs were created in Arc-GIS (Version 10.7). Imagery data were downloaded from open access platform NOAA Data Access Viewer (2018 NOAA Ortho-rectified color Mosaic of Dismal Swamp and Albemarle and Chesapeake Canals, Virginia) on 24 October 2019. Imagery data had horizontal units in meters referenced to the North American Datum of 1983 (NAD83 NSRS2007) projected to UTM Zone 18. Nest data were in the WGS84 projection so we performed a batch re-projection to convert them to the same projection as the imagery data (NAD83 NSRS2007 UTM Zone 18N). Each mapped nest was color-coded by year and labeled with its nest ID. Nests from all years were mapped on the North and South sides of the refuge. The NPM output was scrutinized to determine whether symmetrical (eggs from both clutches reciprocally match one another) or multiple matches could be found among clutches in spatially proximate nests within and between years on each side of the refuge.

Matching was detected first using NPM output from analyses where we ran image data grouped by clutch, using the criterion for inclusion that eggs in one clutch had symmetrical or multiple matches to eggs in another. If a queried clutch had multiple matches to the same other clutch, then the clutch images were inspected visually (denoted Tier I; Fig 2).

We also identified cases where more than one egg in a clutch matched eggs in the same other clutch when the NPM run was batched by clutch without symmetrical matching (e.g. when egg A's closest match is B, but egg B's closest match is egg C). Next, for each clutch, we tallied all the cases identified in the top 8 matches for each egg (in NPM run with each egg in an individual folder) and selected any cases where four or more matched the same other clutch. We used the top 8 matches, because 8 was the average clutch size in the population. We then visually evaluated the inter-clutch matches and selected the match most visually similar to the queried clutch. We used approximate laying dates and final nest fates to further verify these matches as potential renesting attempts, parasitism attempts, or returning breeders. These matches were denoted as Tier IIa for between season matches and Tier IIb for within season matches (Fig 2).

## Results

### Are pattern differences detectable among clutches?

NMDS and PERMANOVA analyses revealed that there were detectable pattern differences among eggs in different clutches and eggs laid by different females for both King Rails and Common Moorhens. PERMANOVA tests for King Rails returned significant differences for individual years 2012, 2014, 2019, 2020, and all years together grouped by clutch (Table 1). Though PERMANOVA tests yielded no significant differences between clutches specifically for 2016, 2017, and all years combined grouped by year, this may have been influenced by the fact that relatively low total numbers of eggs were included from 2016 and 2017. The year 2014 also had a relatively low number of eggs, but it is likely that the 4 clutches included in 2014 were more distinctive than those from 2016 and 2017.

For Common Moorhens, PERMANOVA revealed significant differences between clutches for 1992, 1993, and all three years together (Table 1). There were also significant differences when all years were combined and grouped by either year or clutch and between hens in 1993. $R^2$ values indicate that the greatest differences for all years combined were detected when data were grouped by clutch, secondarily by hen, and lastly by year.

That no significant differences between clutches and between hens were observed for Common Moorhens in 1991 and specifically between hens in 1992 (Table 1) was likely due at least in part to methodological differences in image processing. We partitioned the data according

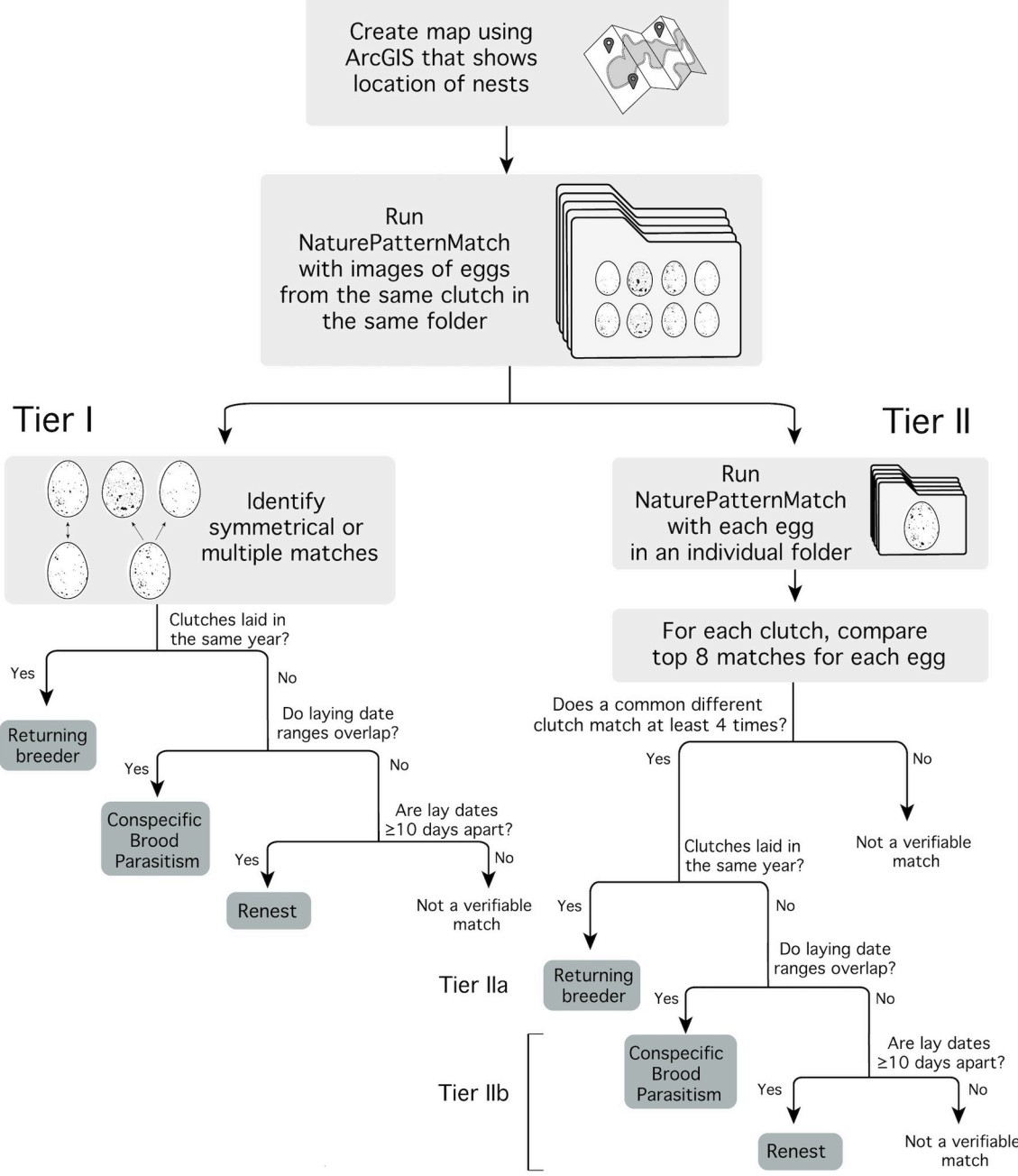

**Fig 2. Process and decision tree for identifying cases of renesting females, likely conspecific brood parasites and renests in King Rails by combining NaturePatternMatch output with spatial and temporal data.**

to year for the Common Moorhens primarily to test for methodological differences in image processing and secondarily according to hen and clutch to see how matching of eggs was affected in the case of multiple clutches per hen. $R^2$ values indicated that the greatest amount of variation in the eggs was explained when clutch was used as the grouping variable rather than hen or year. This suggests that eggs could be more similar within a clutch than eggs from another clutch by the same hen, but only a small number of hens with multiple clutches were included. Alternatively, it could have resulted from differences in lighting or photography.

**Table 1. PERMANOVA output for King Rails and Common Moorhens.**

| Species | Year | Number of clutches | Number of eggs | Number unique hens | $R^2$ | $p$ |
|---|---|---|---|---|---|---|
| King Rail | All together by clutch | 66 | 337 | - | 0.46 | 0.00 |
| Common Moorhen | 1991 by clutch | 7 | 39 | 7 | 0.08 | 0.85 |
| | 1991 by hen | 7 | 39 | 7 | 0.08 | 0.82 |
| | 1992 by clutch | 30 | 200 | 22 | 0.36 | 0.00 |
| | 1992 by hen | 30 | 200 | 22 | 0.01 | 0.12 |
| | 1993 by clutch | 21 | 125 | 18 | 0.40 | 0.00 |
| | 1993 by hen | 21 | 125 | 18 | 0.39 | 0.00 |
| | All together by clutch | 58 | 364 | 38 | 0.39 | 0.00 |
| | All together by hen | 58 | 364 | 38 | 0.29 | 0.00 |
| | All together by year | 28 | 364 | 38 | 0.17 | 0.00 |

PERMANOVA $R^2$ statistics provide information on detectable differences among clutches or hens. Significance was set at 0.05. All years were evaluated together for King Rails and Common Moorhens, and each year was also evaluated individually for Common Moorhens. Grouping was by clutch for King Rails and by clutch or hen for Common Moorhens.

Greater similarity of eggs within clutch was clearly demonstrated by three clutches from the same Common Moorhen female (Fig 3). NPM matched eggs from the same female within and between seasons, despite differences in photography and image processing. Matching was greater within clutch, but additional correct hen matches were detected.

Despite there being statistically significant differences among clutches and hens for Common Moorhens, and within and between years and among clutches for King Rails, it was

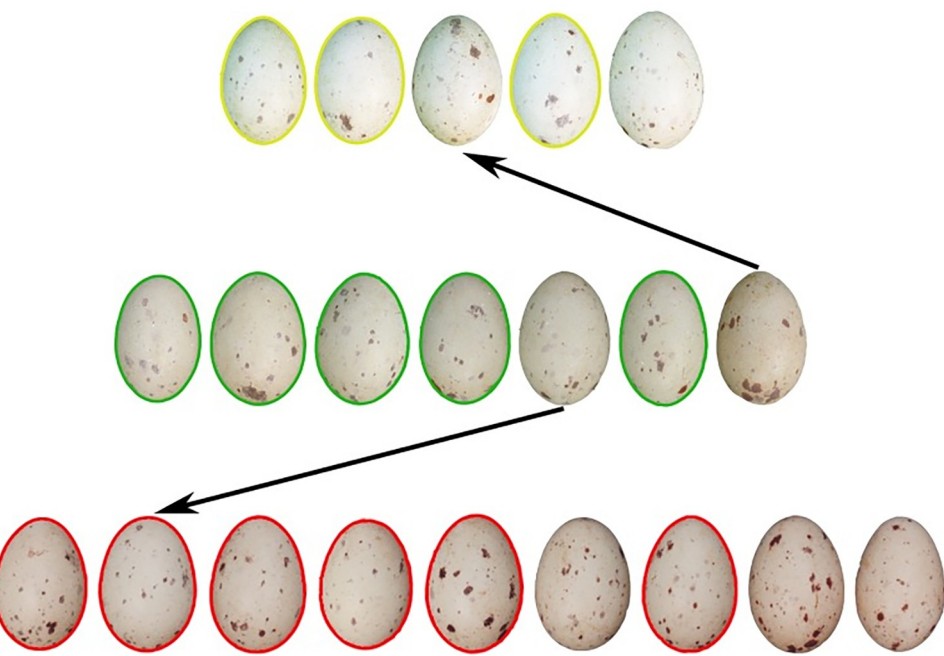

**Fig 3. Three Common Moorhen clutches laid by the same hen in two separate years.** The clutch in the first row was laid in 1992 and the lower two rows are clutches laid in 1993. Colored outlines indicate eggs that matched best with another egg in the same clutch. Arrows indicate NPM matches between clutches. Though a high degree of within-clutch matches suggests a possible confound of photographic differences, matches of eggs with shared maternity both within season and between years occurred in spite of these photographic differences.

challenging to define these pattern differences, and clutches remained difficult to distinguish. Visual inspection suggested that this may have been due in part to pattern variability within clutches. Based on pattern data alone, only a minority of clutches from either species had very distinctive maternal egg signatures.

## Classifying eggs by clutch based on pattern, percent pigmentation, and dimensions

We used linear discriminant analysis to determine whether clutches could be discriminated using a combination of eggshell pattern variables and egg dimensions. The percent of correctly classified eggs was low for both species due to a strong degree of overlap among clutches and hens with some eggs grouping very closely together and others being more variable (Fig 4). The mean percent of eggs correctly classified was higher for Common Moorhens (by clutch = 32.7%, by hen = 32.5%) than for King Rails (by clutch = 20.4%).

Among years, differences in photography in the King Rail study, and in digital image processing of the moorhen clutches could have influenced accuracy of matching. The mean percentage of eggs assigned to their own clutch was slightly elevated for King Rails (29.5%) but was much higher for Common Moorhens (66.7%). The higher classification rate for Common Moorhens when year was used as a grouping variable was likely influenced by the photographic processing methods among years and was therefore unlikely to be biologically meaningful.

## Identifying cases of parasitism, returning breeders, and renesting attempts in King Rails

We mapped and identified spatially proximate King Rail nests across all years. We then looked for symmetrical or repeated NPM matching among spatially proximate nests based on images analyzed for the North and South sides independently. Instances of matching among clutches

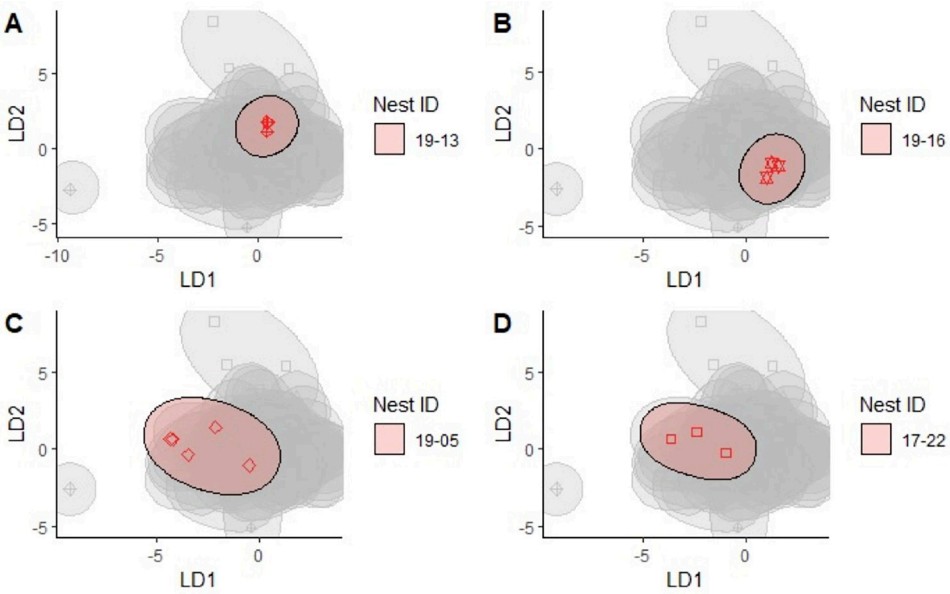

**Fig 4. Linear discriminant analysis of King Rail clutches.** In each plot, a single clutch of eggs is highlighted (red symbols within ellipse) in relation to other clutches (grey ellipses). Graphs A and B are examples of clutches with closely grouped eggs. Graphs C and D show clutches with greater variance.

**Table 2. Putative cases of conspecific brood parasitism, renests and return breeders identified using NPM match data.**

| Tier | Criteria | Nest ID | Distance apart (m) | Estimated first egg date | Nest fate | Last date active | Inferred link |
|------|----------|---------|--------------------|--------------------------|-----------|------------------|---------------|
| Tier I | Physical proximity and/or symmetrical matching | 20–20 | 91 | 5/4/2020 | H/D | 6/2/2020 | CBP |
| | | 20–21 | | 5/3/2020 | H | 6/2/2020 | |
| | | 16–17 | 33 | 4/27/2016 | H | 5/25/2016 | Returning breeder |
| | | 17–22 | | 5/7/2017 | H | 6/8/2017 | |
| | | 19–22 | 61 | 6/3/2019 | H | 6/30/2019 | Returning breeder |
| | | 20–05 | | 4/22/2020 | P | 5/22/2020 | |
| Tier II a | Between year, non-symmetrical matching | 16–29 | 1357 | - | P | 6/7/2016 | Returning breeder |
| | | 19–05 | | 4/28/2019 | H/D | 5/28/2019 | |
| | | 15–25 | 1319 | 6/2/2015 | P/D | - | Returning breeder |
| | | 19–28 | | - | D | 6/23/2019 | |
| Tier II b | Within season, non-symmetrical matching | 19–08 | 851 | 5/9/2019 | H/D | 6/4/2019 | CBP |
| | | 19–15 | | 5/27/2019 | P/H | 6/22/2019 | |
| | | 12–21 | 3305 | 4/21/2012 | P | 5/13-15/2012? | Renesting attempt |
| | | 12–51 | | 6/23/2012 | P | 7/15-17/2012? | |
| | | 20–10 | 440 | Early May | D | 5/17/2020 | CBP |
| | | 20–20 | | 5/4/2020 | H/D | 6/2/2020 | |
| | | 14–03 | 623 | 4/9/2014 | H | 5/12/2014 | CBP |
| | | 14–04 | | 4/9/2014 | H | 5/12/2014 | |

Each pair of nests shown here had at least two eggs match based on two different criteria with NPM output. Tier I nest pairs were identified by physical proximity and/ or symmetrical matching. Tier II nest pairs had at least two eggs in one clutch match eggs in the other and at least four matches between the clutches among the top 8 matches for the query clutch. Tier IIa nests had between-year non-symmetrical matches. Tier IIb nests had within-season non-symmetrical matches. Nest dates (M/D/ YYYY) were also used in evaluating matches identified using NPM and mapped nest locations. Date ranges with a question mark were approximated. Nest fates are abbreviated as H (hatched), D (deserted), or P (depredated). Some nests experienced multiple fates (e.g. partial predation followed by parental desertion, P/D).

were divided into three categories: Tier I based on symmetrical matching or physical proximity and non-symmetrical matching (where eggs in both clutches do not reciprocally match each other), Tier IIa between-year non-symmetrical matching, and Tier IIb within-season non-symmetrical matching.

This method revealed two potential instances of a breeder returning between years. In the first case, eggs laid in nests 19–22 and 20–05 (i.e. likely by the same hen in 2019 and 2020) represented a probable case of site fidelity. The coordinates of these nests were only 61m apart and were situated in a narrow triangular stretch of marsh searched annually that typically had ~3 breeding pairs in it (Tier I in Table 2, Fig 5). The eggs in these clutches featured small maculations largely centered around the large pole of the egg and were similar in shape.

In the second case, clutches 16–17 and 17–22 were inferred to have been laid by the same hen in consecutive years (2016 and 2017). NPM identified matching eggs between nests whose GPS coordinates were only 33m apart (Tier I in Table 2, Fig 5). All the eggs in these clutches were strikingly similar in coloration and shape, with relatively small, dispersed maculations. Spatial proximity, similar appearance and independent matching by NPM strongly suggested a breeding hen with site fidelity.

The first instance of CBP identified occurred with two nests that were only 91m apart (20–20 and 20–21; Tier I in Table 2, Fig 5). The nests were synchronous, with both hatching on the same day. Pattern matching in conjunction with this spatio-temporal pattern suggested a case of brood parasitism by a King Rail hen of her close neighbor.

We queried in NPM individual egg-matching data from the North and South sides of the refuge separately, disregarding finer spatial proximity of nests within those regions. We tested

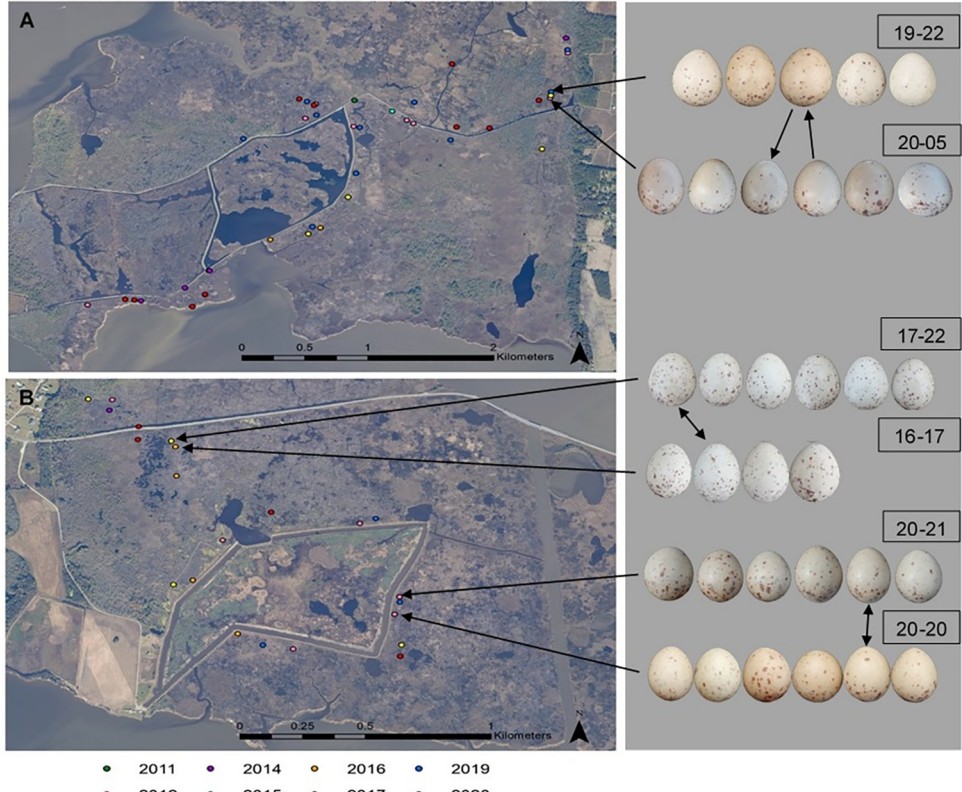

**Fig 5.** Aerial photo of (A) South and (B) North sides of Mackay Island NWR with mapped King Rail nests color-coded by year. According to Tier I criteria, (A) Nests 19–22 and 20–05 and (B) Nests 17–22 and 16–17 each likely represent an instance of a returning female breeder. Whereas Nests 20–20 and 20–21 represent a probable case of conspecific brood parasitism. Double headed arrows among eggs indicate symmetrical pattern matching. Single headed arrows among eggs indicate non-symmetrical pattern matching from query egg to selected match. Imagery data used to create maps available from open access platform NOAA Data Access Viewer (https://coast.noaa.gov/dataviewer/#/).

the sensitivity of our method by increasing the stringency for acceptable matches decrementally from the top 8 (mean clutch size) to the top 5 matches for each egg. Using the higher thresholds for acceptance, only one pair of clutches with matches that we inferred to be a returning breeder dropped out ($n = 65$ clutches total, 40 on South side of refuge, 25 on North side). Thus, considering the top 8 matches for each egg, we identified six instances of best matches between nests (non-symmetrical matching of egg image pairs) (Tier II in Table 2, Fig 6). Non-symmetrical matches between nests from different years that were not spatially proximate to one another were denoted as Tier IIa. Within-season non-symmetrical matches between nests that were not spatially proximate were classified as Tier IIb (Table 2).

Between seasons (Tier IIa criteria) we found two additional clutch matches, three and four years apart, respectively, indicative of likely returned breeding females. The nests were farther apart than the instances identified above (Tier I criteria). Yet, as well as having several non-symmetrical matches identified by NPM, upon visual inspection, eggs in both pairs of clutches were also shaped similarly.

Within season (Tier IIb criteria), we identified three clutch matches indicative of CBP (Fig 6). In each pair of nests, the estimated initiation and hatch dates were within 3 weeks of one another, and matches were based on distinctiveness of specific eggs. Visual inspection revealed

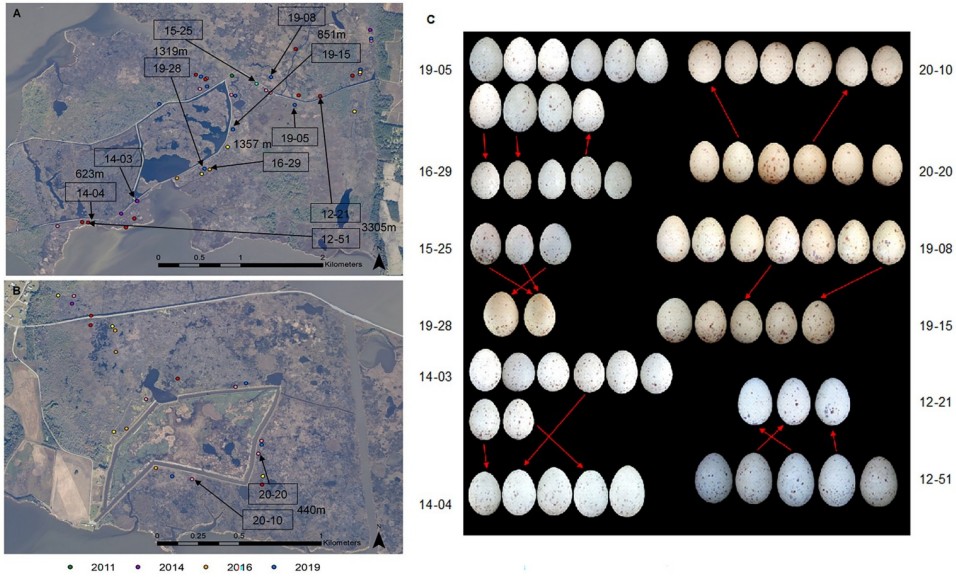

**Fig 6. Spatial relationships among King Rail nests with matching eggs identified by NPM.** Maps of the (A) South and (B) North sides of Mackay Island NWR show King Rail nests color-coded by year. Mapped nest pairs with matching eggs are labeled in adjacent boxes along with the physical distance between them. (C) Corresponding images with matching King Rail eggs from different clutches indicated by arrows based on Tier II criteria. Arrows indicate the direction of non-symmetrical matching from query egg to its NPM-assigned best match. Imagery data used to create maps available from open access platform NOAA Data Access Viewer (https://coast.noaa.gov/dataviewer/#/).

intraclutch variation in egg appearance specifically in nests 14–03, 19–15 and in both 20–10 & 20–20. In one further match inferred to be a within-year renesting attempt, the estimated nest initiation dates were nearly two months apart, and matching results involved each of the eggs in the analysis.

## Discussion

### Pattern-based eggshell matching

NaturePatternMatch, PERMANOVAs, and linear discriminant analyses revealed that King Rails and Common Moorhens appear to have detectable maternal egg signatures and that accurately matching eggs to clutches or hens is possible in some instances. In each species, there was considerable overlap in trait values among clutches that made discrimination difficult based on pattern alone. Yet, some individuals had very distinctive eggs when combinations of traits were considered. Matching using additional variables of egg dimensions and percent pigmentation in conjunction with pattern variables slightly improved discrimination of clutches.

Using NPM, King Rails consistently yielded lower rates of within-clutch matching than Common Moorhens. There was considerable similarity in patterning among eggs in different clutches in both species. Lower egg assignment percentages for King Rails was not entirely unexpected. Though Common Moorhen eggs appear to be more variable overall, King Rails appear to have greater within-clutch variability than Common Moorhens.

Our analyses included images of King Rail clutches spread over more years compared to the moorhens. Given this fragmented sampling regime, there may have actually been few King Rail hens in our sample that laid in more than one nest. Matching might have been lower among years because few hens may be represented in more than one year. Though many of the same

areas were surveyed for nests each year, the entire area searched in any year was only a small proportion of the total available marsh. Further attrition of the dataset of usable images reduced the number of matches among nests we were able to find within and between years. Consistency in photography and image processing should improve matching rates in future studies.

**CBP as an evolutionary driver of variable egg patterns.** King Rails in this study had large home range sizes [40] whereas the Common Moorhens from the British study had small territories and nested at a higher density [43]. High nest density would be expected to be associated with increased levels of CBP, as this conditional reproductive strategy is to some extent opportunistic and density-dependent [44,45]. Nevertheless, high rates of CBP are also correlated with high rates of nest loss [31,43], and nest predation rate was much higher at the King Rail site [47]. Historically higher population size and breeding density in King Rails could have been associated with a higher rate of CBP, which may have selected for egg variability among females. The global population size of King Rails has decreased substantially in recent decades due to range contraction [35]. However, the microsatellite allele frequency distributions in this population are indicative of high genetic diversity and outbreeding [46].

Neither the Common Moorhens [36] nor the King Rails [47] in this study showed well-developed anti-parasite tactics such as recognition and rejection of foreign eggs. Yet, female-specific eggshell signatures may be retained in populations with a history of parasitism. Gomez et al. [48] argued that a Polish population of the confamilial Eurasian Coot (*F. atra*) has retained maternally distinctive eggs though the level of CBP is now negligible due to widely dispersed nests. Using another pattern matching program and a supervised machine learning algorithm, they were able to assign eggs to clutches with an accuracy of 53.3% [48].

**Utility of pattern-based egg matching for rails and other species.** In both Common Moorhens and King Rails, a high degree of phenotypic overlap in egg pattern among clutches laid by different females made discrimination of clutches difficult. Eggs laid by different female Common Cuckoos similarly showed a high degree of phenotypic overlap and were also difficult to classify using automatic clustering methods, such as random forest analysis, or human assessment [29]. Better success in discriminating eggs has been demonstrated in species under strong selection for egg recognition. For example, NPM was used in a comparative study to evaluate egg signatures in four alcid species [15]. Razorbills (*Alle alle*) and Dovekies (*Alca torda*) had very little identity information (weak signatures) in their eggs compared to Common and Thick-billed Murres (*Uria aalge*, *U. lomvia*). Both murre species nest in dense colonies on exposed ledges on cliffs and possess egg recognition [13,49]. In contrast, Razorbills and Dovekies nest in less dense colonies, oftentimes in burrows, and egg recognition has not been noted in either species [14].

NPM evaluates eggshell patterning exclusively. Yet, eggs vary in many other ways including shape, dimensions, and color, all of which are perceived and processed simultaneously by the vertebrate brain [50]. Gómez et al. [48] included 27 variables of eggshell color, spottiness, size, and shape in their classification models. They found that when fewer variables were included correct egg classification decreased, suggesting that multiple eggshell characteristics are used by birds for egg recognition. Due to how the vertebrate visual cortex processes all aspects of egg variation simultaneously, humans may be better at visually discriminating eggs than any program currently available (though opinions on this differ [29,48,51]). Nevertheless, we were able to use matching output from NPM in combination with spatial and temporal data collected in the field to identify probable cases of parasitism and returning breeders that were previously undetected.

## Application of pattern matching to identify maternity in field studies

We mapped the coordinates of King Rail nests and ran pattern matching data from spatially separate areas of the refuge to improve our chances of identifying instances of CBP and within

season re-nesting attempts, and to identify hens that returned as breeders among years. We were able to identify four likely incidences of parasitism, one within-season renesting attempt and four instances of returning breeders, all of which were previously unrecognized. The best match output of NPM was thus useful in a practical way to make inferences about maternal identity of eggs.

In each case of inferred CBP among King Rails, the nests were relatively close to one another (mean inter-nest distance = 501m, range = 91-851m). Females using a mixed strategy of brood parasitism and nesting typically target nearby host nests. Common Moorhen brood parasites were usually from neighboring territories and laid in their own nest immediately after laying parasitically [38]. Moreover, brood parasitism is only successful when parasitic eggs hatch at the same time as host eggs, and local conditions may favor synchronized breeding attempts among neighbors [43]. Estimated mean King Rail territory sizes were large (~3 ha) [40]. Brood rearing King Rails in this population are also known to travel significant distances away from their nests (average maximum distance: ~600 ± 200 m) [40]. It is therefore conceivable that female King Rails travel comparable distances to renest or to parasitize another nest.

Four cases of among-year site fidelity of breeding females were identified based on pattern matching of eggs in conjunction with proximity of mapped nests. Two of these occurred in sequential years, whereas the others were three and four years apart, respectively, well within a reasonable timeframe for an adult King Rail to survive and breed again, even given a high estimated rate of adult predation [52]. A single other case of a female nesting in a territory adjacent to the one she nested in the year before had been documented in this population previously, discovered by parentage analysis based on microsatellite genotyping [46]. While site fidelity among territorial males had been reported in this resident population previously, based on radiotelemetry [40], this new evidence supports a level of nest site fidelity among breeding females as well.

An alternative explanation for these matches is the possibility that different hens laid the eggs but they were close relatives (mother-daughter, sisters) and shared heritable characteristics of their eggshell patterns. Eggshell patterning has been found to be a female sex-linked heritable trait in Great Tits, where eggs of daughters match best with their mothers and maternal grandmothers [23]. Similarly, a study of Japanese Quail (*Coturnix japonica*) suggested intermediate or high levels of heritability in eggshell coloring and patterning [53]. While we cannot discount the possibility of closely related King Rail hens, testing this hypothesis will have to wait until we have a better understanding of recruitment patterns and rates in this population.

NPM and other egg matching programs have been shown to match eggs correctly without considering genetic information [10,29]. While it will be important in the future to validate studies using pattern matching programs to characterize eggs of different species with genetic parentage and pedigree data, image matching techniques offer an alternative minimally invasive sampling method. Collecting genetic samples through capture can be stressful for the animals, and demanding of limited field resources, whereas photographing clutches and evaluating the egg images is a less invasive way to monitor breeders. This has important implications for working with secretive species and species of conservation concern. Pattern analysis techniques should continue to be investigated as a potential monitoring method for species laying maculated eggs.

## Acknowledgments

We thank Mike Hoff and his staff at Mackay Island NWR and the Department of Biology of East Carolina University for their support. We are grateful to field assistants Kristen Orr, Jacob

Goldman, Nathaniel Watkins, Robert Dcamp, Ben Decker, AG Sweany, and Susannah Halligan for their hard work in the field, and to committee members Michael Brewer, Michael McCoy, and Kyle Summers for their suggestions. Valuable help was provided in statistics by Cody Kent and in graphic design by Kristen Orr.

## Author Contributions

**Conceptualization:** Emily W. Johnson, Susan B. McRae.

**Data curation:** Emily W. Johnson.

**Formal analysis:** Emily W. Johnson.

**Funding acquisition:** Emily W. Johnson, Susan B. McRae.

**Investigation:** Emily W. Johnson, Susan B. McRae.

**Methodology:** Emily W. Johnson, Susan B. McRae.

**Project administration:** Susan B. McRae.

**Resources:** Susan B. McRae.

**Supervision:** Susan B. McRae.

**Validation:** Emily W. Johnson, Susan B. McRae.

**Visualization:** Emily W. Johnson.

**Writing – original draft:** Emily W. Johnson.

**Writing – review & editing:** Emily W. Johnson, Susan B. McRae.

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
