## [Decision Letter · Decision Letter 0]

8 Oct 2021

PONE-D-21-28285Interclutch variability in egg characteristics in two species of rail: is maternal identity encoded in eggshell patterns?PLOS ONE

Dear Dr. McRae,

Thank you for submitting your manuscript to PLOS ONE. After careful consideration, we feel that it has considerable merit but does not fully meet PLOS ONE’s publication criteria as it currently stands. Therefore, we invite you to submit a revised version of the manuscript that addresses the points raised during the review process.  The reviewers' comments can be found below.  The changes required are relatively minor.

We look forward to receiving your revised manuscript.

Sincerely,

Charles R. Brown

Academic Editor

PLOS ONE

Journal Requirements:

“This project was made possible by funding from the U.S. Fish and Wildlife Service and the Association of Field Ornithologists. V”

We note that you have provided funding information within the Acknowledgements Section. Please note that funding information should not appear in the Acknowledgments section or other areas of your manuscript. We will only publish funding information present in the Funding Statement section of the online submission form.

“This study was conducted with support from the U.S. Fish and Wildlife Service, Refuge System Inventory and Monitoring program through a Piedmont South Atlantic Coast Cooperative Ecosystems Studies Unit (http://www.cesu.psu.edu/unit_portals/PSAC_portal.htm) agreement to SBM (F19AC00629), and an E. Alexander Bergstrom Memorial Research Award from the Association of Field Ornithologists (https://afonet.org/grants-awards/bergstrom/) to EWJ. The funders had no role in study design, data collection and analysis, decision to publish, or preparation of the manuscript.”

4. We note that you have included the phrase “unpublished data” in your manuscript. Unfortunately, this does not meet our data sharing requirements. PLOS does not permit references to inaccessible data. We require that authors provide all relevant data within the paper, Supporting Information files, or in an acceptable, public repository. Please add a citation to support this phrase or upload the data that corresponds with these findings to a stable repository (such as Figshare or Dryad) and provide and URLs, DOIs, or accession numbers that may be used to access these data. Or, if the data are not a core part of the research being presented in your study, we ask that you remove the phrase that refers to these data.

5. We note that Figure 3 and 4 in your submission contain satellite images which may be copyrighted. All PLOS content is published under the Creative Commons Attribution License (CC BY 4.0), which means that the manuscript, images, and Supporting Information files will be freely available online, and any third party is permitted to access, download, copy, distribute, and use these materials in any way, even commercially, with proper attribution. For these reasons, we cannot publish previously copyrighted maps or satellite images created using proprietary data, such as Google software (Google Maps, Street View, and Earth). For more information, see our copyright guidelines: http://journals.plos.org/plosone/s/licenses-and-copyright.

 a. You may seek permission from the original copyright holder of Figure 3 and 4 to publish the content specifically under the CC BY 4.0 license. 

Reviewers' comments:

Reviewer's Responses to Questions

**Comments to the Author**

1. Is the manuscript technically sound, and do the data support the conclusions?

Reviewer #1: Yes

Reviewer #2: Yes

2. Has the statistical analysis been performed appropriately and rigorously? 

Reviewer #1: Yes

Reviewer #2: Yes

3. Have the authors made all data underlying the findings in their manuscript fully available?

Reviewer #1: Yes

Reviewer #2: Yes

4. Is the manuscript presented in an intelligible fashion and written in standard English?

Reviewer #1: Yes

Reviewer #2: Yes

5. Review Comments to the Author

Reviewer #1: This paper is very well written and methodologically sound. It makes novel contributions and presents promising results regarding the use of pattern matching and statistical analyses to assess intra- vs inter-clutch variation in multi-year samples of eggs of two bird species. The discussion of results also describes how such analyses may lead to the detection of potential instances of conspecific brood parasitism, re-nests and return breeders without the need for genetic or more detailed observational studies.

Methodologically the paper is sound and generally easy to follow, although the sequence of steps described in some sections, such as "Spatial and temporal analysis of King Rail matching results", could in my opinion be more clearly, concisely and unambiguously conveyed by replacing or augmenting the prose with a diagram such as a flow chart or some pseudo-code.

Implementation decisions, such as "we used the top 8 matches", are justified in the text, but it would be good to conduct at least some rudimentary sensitivity analysis to determine whether the analyses that were performed are highly dependent on such choices. A little more analysis regarding the impact of confounding factors such as differences in photographic and image processing across the different years, as well as differences in sample sizes, would also have strengthened the assertions made in the paper.

Figures 3 and 4 are very useful in clarifying the spatial and pattern similarity linkages, but a few more detailed exemplars of matched egg images would have been helpful.

Reviewer #2: This is well-conceived research project about an interesting topic. It was cleverly and meticulously carried out, with both the methods and statistical analyses thoroughly described. The comparison between moorhens and rails added substantially to the authors arguments. My only suggestion is to explain symmetrical and non-symmetrical matching the first time it is mentioned in the text--it took me a couple of read-throughs to understand what that meant.

6. PLOS authors have the option to publish the peer review history of their article (what does this mean?). If published, this will include your full peer review and any attached files.

Reviewer #1: No

Reviewer #2: No

---

## [Author Response · Author response to Decision Letter 0]

21 Nov 2021

Responses to reviews below

done 

“This project was made possible by funding from the U.S. Fish and Wildlife Service and the Association of Field Ornithologists. V”

We note that you have provided funding information within the Acknowledgements Section. Please note that funding information should not appear in the Acknowledgments section or other areas of your manuscript. We will only publish funding information present in the Funding Statement section of the online submission form.

“This study was conducted with support from the U.S. Fish and Wildlife Service, Refuge System Inventory and Monitoring program through a Piedmont South Atlantic Coast Cooperative Ecosystems Studies Unit (http://www.cesu.psu.edu/unit_portals/PSAC_portal.htm) agreement to SBM (F19AC00629), and an E. Alexander Bergstrom Memorial Research Award from the Association of Field Ornithologists (https://afonet.org/grants-awards/bergstrom/) to EWJ. The funders had no role in study design, data collection and analysis, decision to publish, or preparation of the manuscript.”

We have removed the sentence in the Acknowledgments. We do not wish to amend the funding statement.

The full dataset can be found at https://doi.org/10.5061/dryad.6q573n60j

We have not yet received accession numbers from DataDryad because the datasets are still being reviewed by the curators. It was a challenge to get the revisions and data curation completed with my co-author over distance. It was a busy hybrid academic term for me, and we completed these revisions while Emily was translocating to a new position in the Florida Keys. We wanted to get the revisions back to you by the deadline, but I will send along the accession numbers as soon as they are available. 

4. We note that you have included the phrase “unpublished data” in your manuscript. Unfortunately, this does not meet our data sharing requirements. P LOS does not permit references to inaccessible data. We require that authors provide all relevant data within the paper, Supporting Information files, or in an acceptable, public repository. Please add a citation to support this phrase or upload the data that corresponds with these findings to a stable repository (such as Figshare or Dryad) and provide and URLs, DOIs, or accession numbers that may be used to access these data. Or, if the data are not a core part of the research being presented in your study, we ask that you remove the phrase that refers to these data.

The reference ‘Clauser AJ, McRae SB. King Rails (Rallus elegans) vary building effort and nest height in relation to water level. Waterbirds. 2016;39(3): 268–76’ reports the nest predation rate over a 2-year period of study, which is representative of the population. We now include a reference to that paper. We have also replaced the citation of the thesis by C.L. Brackett and replaced it with other citations, because it is not available online. 

5. We note that Figure 3 and 4 in your submission contain satellite images which may be copyrighted. All PLOS content is published under the Creative Commons Attribution License (CC BY 4.0), which means that the manuscript, images, and Supporting Information files will be freely available online, and any third party is permitted to access, download, copy, distribute, and use these materials in any way, even commercially, with proper attribution. For these reasons, we cannot publish previously copyrighted maps or satellite images created using proprietary data, such as Google software (Google Maps, Street View, and Earth). For more information, see our copyright guidelines: http://journals.plos.org/plosone/s/licenses-and-copyright.

 a. You may seek permission from the original copyright holder of Figure 3 and 4 to publish the content specifically under the CC BY 4.0 license. 

We only used open source maps. We have added language to the methods to clarify the origin of these maps. All imagery to create maps was accessed from the open access platform NOAA Data Access Viewer (https://coast.noaa.gov/dataviewer/#/) and the exact imagery file (2018 NOAA Ortho-rectified color Mosaic of Dismal Swamp and Albemarle and Chesapeake Canals, Virginia) name has been provided in the text, as well as the date the imagery file was originally downloaded by the authors (24 October 2019). 

N/A

Reviewers' comments:

Reviewer's Responses to Questions

Comments to the Author

1. Is the manuscript technically sound, and do the data support the conclusions?

Reviewer #1: Yes

Reviewer #2: Yes

2. Has the statistical analysis been performed appropriately and rigorously? 

Reviewer #1: Yes

Reviewer #2: Yes

3. Have the authors made all data underlying the findings in their manuscript fully available?

Reviewer #1: Yes

Reviewer #2: Yes

4. Is the manuscript presented in an intelligible fashion and written in standard English?

Reviewer #1: Yes

Reviewer #2: Yes

5. Review Comments to the Author

Reviewer #1: This paper is very well written and methodologically sound. It makes novel contributions and presents promising results regarding the use of pattern matching and statistical analyses to assess intra- vs inter-clutch variation in multi-year samples of eggs of two bird species. The discussion of results also describes how such analyses may lead to the detection of potential instances of conspecific brood parasitism, re-nests and return breeders without the need for genetic or more detailed observational studies.

Thank you!

Methodologically the paper is sound and generally easy to follow, although the sequence of steps described in some sections, such as "Spatial and temporal analysis of King Rail matching results", could in my opinion be more clearly, concisely and unambiguously conveyed by replacing or augmenting the prose with a diagram such as a flow chart or some pseudo-code.

We have added a flow chart (new Fig 2) to facilitate understanding of this section.

Implementation decisions, such as "we used the top 8 matches", are justified in the text, but it would be good to conduct at least some rudimentary sensitivity analysis to determine whether the analyses that were performed are highly dependent on such choices. 

In our criteria for including nest pairs with non-symmetric matching, we examined the 8 top pattern matches for each egg due to that being the mean clutch size in the king rail population. We conducted a sensitivity analysis to determine how increasing the stringency by limiting the number of top matches further would have affected the results. When we reduced the number of top matches we included decrementally from 8 to 5, only one pair of clutches with matches that we inferred to be a returning breeder dropped out (N=65 clutches, 40 on the South side and 25 on the North side; table of results below). We have added a couple of sentences to the Results to explain this.

Number of top matches included Total (%) clutches with match % Returning breeder % CBP % Renesting

5 5 (7.7%) 1.5% 4.6% 1.5%

6 5 (7.7%) 1.5% 4.6% 1.5%

7 5 (7.7%) 1.5% 4.6% 1.5%

8 6 (9.2%) 3.1% 4.6% 1.5%

A little more analysis regarding the impact of confounding factors such as differences in photographic and image processing across the different years, as well as differences in sample sizes, would also have strengthened the assertions made in the paper.

Please see next answer. 

Figures 3 and 4 are very useful in clarifying the spatial and pattern similarity linkages, but a few more detailed exemplars of matched egg images would have been helpful.

Thank you for the positive feedback. We have now also added a figure (Fig 3) illustrating three clutches of Common Moorhen eggs laid by the same female, showing multiple within-clutch best matches, as well as between-clutch matches within- and between years. This illustration and description helps to address the previous comment also in that it shows how eggs within the same photograph were more likely to have other eggs in the same clutch and photograph as their top matches. We have added an explanation of this in the text.

Reviewer #2: This is well-conceived research project about an interesting topic. It was cleverly and meticulously carried out, with both the methods and statistical analyses thoroughly described. The comparison between moorhens and rails added substantially to the authors arguments. My only suggestion is to explain symmetrical and non-symmetrical matching the first time it is mentioned in the text--it took me a couple of read-throughs to understand what that meant.

Thank you for your comments and for pointing out this omission. We have added some language earlier in the methods (lines 335 and 344) to explain symmetrical and non-symmetrical matching.

6. PLOS authors have the option to publish the peer review history of their article (what does this mean?). If published, this will include your full peer review and any attached files.

Do you want your identity to be public for this peer review? For information about this choice, including consent withdrawal, please see our Privacy Policy.

Reviewer #1: No

Reviewer #2: No

---

## [Editor Report · Decision Letter 1]

13 Dec 2021

Interclutch variability in egg characteristics in two species of rail: is maternal identity encoded in eggshell patterns?

PONE-D-21-28285R1

Dear Dr. McRae,

We’re pleased to inform you that your manuscript has been judged scientifically suitable for publication and will be formally accepted for publication once it meets all outstanding technical requirements.

Sincerely,

Charles R. Brown

Academic Editor

PLOS ONE
---

## [Editor Report · Acceptance letter]

17 Dec 2021

PONE-D-21-28285R1 

Interclutch variability in egg characteristics in two species of rail: Is maternal identity encoded in eggshell patterns? 

Dear Dr. McRae:

I'm pleased to inform you that your manuscript has been deemed suitable for publication in PLOS ONE. Congratulations! Your manuscript is now with our production department. 

Kind regards, 

on behalf of

Dr. Charles R. Brown 

Academic Editor

PLOS ONE